# COVID-19 Prophylaxis Efforts Based on Natural Antiviral Plant Extracts and Their Compounds

**DOI:** 10.3390/molecules26030727

**Published:** 2021-01-30

**Authors:** Oksana Sytar, Marian Brestic, Shokoofeh Hajihashemi, Milan Skalicky, Jan Kubeš, Laura Lamilla-Tamayo, Ulkar Ibrahimova, Sayyara Ibadullayeva, Marco Landi

**Affiliations:** 1Department of Plant Physiology, Slovak University of Agriculture, A. Hlinku 2, 94976 Nitra, Slovakia; 2Department of Plant Biology, Institute of Biology, Kiev National, University of Taras Shevchenko, Volodymyrska, 64, 01033 Kyiv, Ukraine; 3Department of Botany and Plant Physiology, Faculty of Agrobiology, Food and Natural Resources, Czech University of Life Sciences Prague, Kamycka 129, 16500 Prague, Czech Republic; skalicky@af.czu.cz (M.S.); kubes@af.czu.cz (J.K.); lamilla_tamayo@af.czu.cz (L.L.-T.); 4Plant Biology Department, Faculty of Science, Behbahan Khatam Alanbia University of Technology, 47189-63616 Khuzestan, Iran; hajihashemi@bkatu.ac.ir; 5Institute of Molecular Biology and Biotechnology, Azerbaijan National Academy of Sciences, Matbuat Avenue 2A, Az 1073 Baku, Azerbaijan; u.ibrahimova@yahoo.com (U.I.); ibadullayeva.sayyara@mail.ru (S.I.); 6Department of Agriculture, Food and Environment, University of Pisa, 56126 Behbahan, Italy

**Keywords:** COVID-19, coronaviruses group, biological active compounds, plant chemo-diversity

## Abstract

During the time of the novel coronavirus disease 2019 (COVID-19) pandemic, it has been crucial to search for novel antiviral drugs from plants and well as other natural sources as alternatives for prophylaxis. This work reviews the antiviral potential of plant extracts, and the results of previous research for the treatment and prophylaxis of coronavirus disease and previous kinds of representative coronaviruses group. Detailed descriptions of medicinal herbs and crops based on their origin native area, plant parts used, and their antiviral potentials have been conducted. The possible role of plant-derived natural antiviral compounds for the development of plant-based drugs against coronavirus has been described. To identify useful scientific trends, VOSviewer visualization of presented scientific data analysis was used.

## 1. Introduction

COVID-19 was characterized and announced as a pandemic on 11 March 2020. The novel coronavirus was first reported in Wuhan, Chinaand spread around the world from there. On 12 February 2020, the WHO announced a name for the new coronavirus disease: COVID-19, which became the fifth recorded pandemic since the 1918 influenza outbreak [1,2,3]. Across the world, governments supported safety by minimizing human contact through country-wide lockdowns of public places, limiting gatherings, imposing quarantines. and mandating mask wearing in public and social distancing. These measures did help to slow the spread of SARS-CoV-2 [4,5]. Major efforts were undertaken to develop vaccines and to find effective drugs and therapies to reduce infectivity. At time of writing, there are more than 50 vaccine candidates under development [6,7], and the WHO has already listed the Pfizer-BioNTech vaccine for emergency use. At the same time, antiviral extracts and compounds with the potential to limit virus transmission or block infection are also being developing [8]. Plant production technology is being adopted to create vaccines and inexpensive antiviral proteins [9]. Different kinds of plants such as turnip, potato, tobacco, and others have been used for vaccine production [10].

At the same time, it is important to educate people in the use of plants that help to support the immune system, and to establish research programs to develop functional foods such as probiotics that can suppress viral infection. Naja and Hamdeh (2020) described a multi-level action plan to support optimal nutrition at the individual, societal, and national levels during the pandemic [11]. The main goal was to maintain a healthy diet, but this must be supported by creating a comprehensive scientific database on herbal products and plant derivatives, which can be used to prevent or treat illness in many different countries. The types of bioactive compounds from crops and herbs and the mechanism of their support of the human immune system against infections have been thoroughly discussed, along with the absence of evidence for SARS-CoV-2 transmission through the food chain [12,13].

Plants represent a major source of chemo-diversity on the planet, and it is likely that some safe and effective plant compounds can be found that could help to protect human lives from the devastation of COVID-19. In the scientific databases, there are huge numbers of research articles about the antiviral, antifungal, antibacterial, antiviral, and anthelmintic activities of medicinal herbs and crops with different ethnobotanical background [14,15,16,17]. Connections between the health effects of medicinal plants and presence of specific secondary metabolites which can be responsible for some effects are also described. It was suggested that terpenoids, alkaloids, stilbenes, and flavonoids are the main biological active compounds for the development of antiviral plant-based drugs [18]. To search antiviral plant-based drugs among such a large quantity of scientific data, however, caution must be used to guard against misinformation and bogus recommendations. 

As a complement to the recommended COVID-19 prophylaxis, the evaluation of the effectiveness of plant extracts of different medicinal herbs and crops and natural antiviral compounds could be included in randomized controlled testing of large populations. Therefore, the present review presents detailed characteristics of medicinal plants and crops based on their ethnobotanical background, the plant part used, their antiviral potential, and already known plant-based antiviral compounds. To develop further scientific trends for the prophylaxis of COVID-19, VOSviewer visualization was used.

## 2. Results and Discussion

### 2.1. Ethnobotanical Background of Plants with Antiviral Potential

During the pandemic, many scientists have concentrated on how best to take care of the population before a vaccine becomes available. One of their goals is to develop an efficient viral inactivation system by exploiting naturally occurring antiviral compounds from medicinal plants. One example of this strategy would be to incorporate such a compound into nanofiber respiratory masks [5]. The contemporary clinical efficacy and safety profile of medications such as hydroxychloroquine and half synthetic antibiotic azithromycin against COVID-19 have been described as well [19]. Another strategy is to enhance people’s antiviral immune response through a nutritious diet including plant-derived supplements characterized by high antiviral potential to minimize the risk of SARS-CoV-2 and similar infections. At the same time, natural antiviral plant compounds can be used to develop antiviral plant drugs. Several studies on the antiviral potential of plant extracts have been conducted using in vitro model systems of cultured Vero cells or human cell lines as a pre-clinical stage of research. During the SARS-CoV-2 pandemic, researchers have been taking advantage of molecular docking models for testing chemical libraries for bioactive compounds [20]. These new techniques have been utilized together with pharmacological network analysis to characterize biologically active compounds from fruits of *Juniperus communis* and herbs such as *Thymus vulgaris*, *Curcuma longa*, *Rosmarinus officinalis*, *Ocimum basilicum*, *Melissa officinalis*, and *Mentha piperita* [21]. It has been shown that quercetin from onions (*Allium cepa*), apples (*Malus domestica*), green tea (*Camellia sinensis*) and buckwheat (*Fagopyrum esculentum*) can inhibit the 6LU7 and 6Y2E proteases of SARS-CoV-2 by binding to them [20,22]. The latest information from the literature on plants with antivirus potential is presented in Table 1. This table lists the medicinal plants as therapeutic tools to suppress various viral agents such as Herpes simplex virus type 1 and 2 (HSV-1, HSV-2), human immunodeficiency virus (HIV), SARS-CoV-2, HCoV-NL63, influenza A, and others causing diseases such as oral and genital herpes, AIDS, SARS, or flu.

Viruses can develop resistance through mutation to current antimicrobial agents, and this increases the need for the discovery and development of new effective compounds against old and new viral infections, especially against SARS-CoV-2. The source of such drugs is often a library of plant-derived compounds with antiviral properties and various mechanisms of action. Secondary metabolites such as terpenes, flavonoids, alkaloids, saponins, and stilbenes have been characterized through antiviral activity assays [23]. Some plants containing compounds that inhibited viruses were identified from Gillan’s plants by Iranian scientists, and specific alkaloids were extracted from them such as trshvash, chuchaq, cote d’couto and khlvash in Iran. The names of the plants were those used in particular regions of Iran and the Latin names were not given [24]; however, it seems that they correspond with *Oxalis corniculata*, *Eryngium planum*, and *Ziziphora persica*. Some people in Iran drink a tea made from a mixture of *Stachys schtschegleevi* and *Origanum vulgare* to prevent infection, while those infected with coronavirus are told to drink *Echium amoenum* tea daily. Local healers encourage people to gargle with an extract of *Rhus coriaria* or *Myrtus communis* along with other practices such as hand washing and wearing masks to block the virus from entering the nose and respiratory system. Currently in China, almost 85% of patients treated for COVID-19 are receiving treatment with traditional Chinese medicine based on the observed antiviral effects of plant-derived compounds [25,26].

The literature review, performed to create a scientific database of plants with antiviral potential, obtained results from a search of PubMed and Google Scholar combined with some of the primary traditional Persian manuscripts on medicine, including the book of AlHavi, the Canon of Medicine, Zakhireh-iKharazmshahi, Qarabadine-kabir, Tohfat ol Moemenin, and Makhzan-ol-advieh. The search was performed with terms for medicinal plants used in treating respiratory system disorders [27,28]. Nearly twenty medicinal plants containing mucilage were found [27]. The descriptions of plants with antiviral activity that could be useful for future studies against COVID-19 and other diseases are presented in Table 1. The literature analysis, which was performed to create a scientific database of plants with antiviral potential (Table 1), obtained results from a search of PubMed, Scopus and Web of Knowledge databases combined with some of the primary traditional Azerbaijani manuscripts on herbal medicine.

### 2.2. Plant-Derived Antiviral Compounds Against Coronavirus

The novel coronavirus is a respiratory virus that spreads through droplets of saliva or discharge from the nose. The main knowledge, with the parallel development of vaccines, is to avoid any contact with an infected person’s breath or sneezes [127]. At the same time, identifying novel antiviral drugs is of critical importance and natural products are an excellent source for such discoveries [128].

The plant metabolites with antiviral activity can kill viruses and prevent the infection of the respiratory system [129,130]. The coronavirus genome is a single-stranded RNA, comprising about 30,000 nucleotides encoding four structural proteins—a membrane protein, a spike protein, an envelope protein, and a nucleocapsid protein—and some nonstructural proteins. The nucleocapsid is a shell made of protein that surrounds the single-stranded, positive-strand RNA genome [131]. The N protein coats the RNA and allows the virus to enter and hijack human cells, turning them into virus factories. The mechanism of formation of the virus structure is one of the important research areas for the development of drugs to prevent virus particles from binding to human cells and infecting them. Plant metabolites are highly antioxidant and can be used for the development of plant-based drugs or to support the immune system [14,19].

The source of such plant-based drugs is often a chemical library of plant-derived compounds with antiviral properties and a variety of mechanisms of action. Plant natural products or extracts have been used in folk medicine for hundreds of years to treat viral diseases [132]. The market for herbal supplements with specific nutraceutical properties is huge [133,134,135], and with the pandemic threatening people’s lives and livelihoods, it is logical select natural products with antiviral potential. Some natural compounds such as lycorine, homoharringtonine, silvestrol, ouabain, tylophorine, and 7-methoxycryptopleurine have been reported to have antiviral activity at nanomolar concentrations and may lead to future antiviral drug development [132]. Many natural products possess anti-coronavirus potential, and these should be studied as potential dietary supplements for reducing infectivity and modulating the immune response.

#### 2.2.1. Zoonotic Interventions

Regarding the zoonotic origins of some viruses, including SARS-CoV-2, plants and their products should also be tested for animal treatment and promising compounds could be further investigated for human application. Lelešius et al. (2019) examined extracts of different medicinal herbs and found that *Mentha piperita*, *Thymus vulgaris* and *Desmodium canadense* extracts were the most effective against avian infectious bronchitis virus prior to and during infection [53]. Kaempferol and its glucoside derivatives (rhamnose), the kaempferol glycosides afzelin, juglanine and tiliroside from the holly oak, *Quercus ilex* L., were also effective against the 3a channel protein of a coronavirus [136]. Holm oaks (*Quercus Ilex* L.), oleanders (*Nerium Oleander* L.), and wild olive trees (*Olea europaea* L.) are typical Mediterranean plants presented in Italian forested areas, which are suggested to have immunoprotecting potential and have shown lower COVID-19 mortality rates for the population [137]. 

#### 2.2.2. Antiviral Flavonoids

Naturally occurring flavonoid compounds have high antiviral potential. For example, the flavonoid scutellarein, from the root of *Scutellaria baicalensis* (Lamiacaea), has been shown to inhibit the nsP13 helicase of SARS-CoV-2 by altering its ATPase activity [110]. The pharmacology research has shown the potential therapeutic effects of baicalin and baicalein, which are other specific flavone glycosides of *Scutellaria baicalensis*, in response to COVID-19. The exact therapeutic effects of *Scutellaria baicalensis* extract still needs to be determined in clinical trials [138]. It is important to remember that the use of extracts is complicated by the presence of multiple compounds. The antiviral and antimicrobial capacities of separate biologically active compounds may be different from their effects in extracts, which can be additive or synergistic, or even antagonistic [139,140]. For example, after removal of the tannins from the aqueous extract of *Euphorbia hirta* L., the inhibition of viral replication was markedly decreased. It was concluded that the tannins were likely responsible for the high antiretroviral activity [141]. Studies using protein-molecular docking with network pharmacology analysis were able to identify and characterize other bioactive compounds from the fruits of *Juniperus communis* and the brown alga, *Ecklonia cava* [76,79]. Here, quercetin isolated from *E. cava* was effective in treating respiratory diseases and could be used to mitigate the airway damage from SARS-CoV-2 infection directly in the respiratory system.

#### 2.2.3. Antiviral Terpenoids

Terpenoids constitute a large group of secondary metabolites with a wide spectrum of structures and effects, including antiviral properties. Some of these compounds are relatively simple, such as monoterpenes from *Lamiaceae* plants in Table 1 [142]. More complex molecules were isolated and identified from species of the genus *Bupleurum*, which is widespread in the old world and an important herb in Chinese traditional medicine. Antiviral effects of its triterpenoid saikosaponins were studied on a human fetal lung fibroblast model. It was found that saikosaponins were active in the early stage of HCoV-22E9 infection, preventing viral attachment and penetration [143]. Other identified active compounds from this plant included the above-mentioned antioxidant flavonoids such as quercetin, isorhamnetin, narcissin, rutin, eugenin, and saikochrome A, which may have anti-inflammatory effects [45]. The oleanane triterpenes in ethanolic extracts of *Camellia japonica* flowers were shown to possess significant antiviral activity against the PEDV coronavirus in research with a Vero cell model. Inhibitory effects on key gene and protein synthesis during PEDV replication have been shown as well [46]. *Camellia japonica* does not produce purine alkaloids, but it does contain the flavonoids quercetin, kaempferol and apigenin [144,145,146]. A screening study of antiviral action of herbal extracts in 1979 found that a triterpenoid saponin from licorice (*Glycyrrhyza glabra)* roots, glycyrrhyzic acid, was active against viruses [147]. This plant is native to the Mediterranean region, Iran-Turan, and Azerbaijan, where it has been used in folk medicine for many years. The latest studies confirmed the antiviral activity towards HSV-1, Epstein–Barr virus, human cytomegalovirus, and RNA viruses such as influenza A virus (IAV), H5N1, and H1N1, and the immunomodulation capacity of its extract [64,148,149] inhibits the PLpro and 3CLpro targets, which are known to be essential for viral replication [16,18,150]. The rational supporting combinations of glycyrrhizin with tenofovir and (hydroxy)chloroquine (two drugs active against SARS-CoV-2) are discussed but need more clinical studies [151].

In another study using molecular docking, researchers examined 171 essential oils (see Table 1) in connection with specific enzymes of SARS-CoV-2, such as the main protease, endoribonuclease, and RNA-dependent RNA polymerase. However, the most promising compounds such as isomers of farnesene and (*E*,*E*)-farnesol did not display strong docking activity. Despite this, the authors suggested that they possessed a hypothetical synergistic effect in the whole extract [152].

*Cannabis sativa* originated in central Asia and has been used for treating illness for more than 5000 years [152]. The phytocannabinoid, cannabidiol (CBD), was discovered in *C. sativa* in 1940 and makes up about 40% of the extract [153]. CBD has been shown to be a modulator of angiotensin-converting enzyme II (ACE2) expression in COVID-19 target tissues [154]. Down-regulation of ACE2, which is associated with receptor-mediated entry into human lung epithelial cells, may provide a reasonable strategy for reducing COVID-19 severity. Researchers in Canada have developed over 800 new *C. sativa* lines and extracts with high-CBD content for antiviral testing. It was also confirmed that some *C. sativa* extracts down-regulated another protein required for SARS-CoV-2 entry into host cells, the serine protease, TMPRSS2 [154]. While preclinical studies encourage the potential effectiveness of CBD in viral diseases such as Kaposi sarcoma, hepatitis C and SARS-CoV-2, clinical evidence is still lacking [155].

South America and Africa have a very long history of utilizing native plants in traditional medicine, which has played an important role in the health of the population. There may be some evidence for this assertion; an extract from the bark of *Ampelozizyphus amazonicus* Ducke was tested and found to have immunomodulatory and anti-inflammatory activities [36]. In Brazil, this plant is widely used to prevent malaria and it is known to contain triterpenic saponins, averaging about 48% of the dry weight [156]. *Artemisia* species native to Madagascar have been discovered to possess antimalarial effect as well, probably thanks to sesquiterpenic lacton artemisinin [157], and they have antiviral and immunomodulatory potential [41,98]. Beside *Artemisia*, other plants from Madagascar were previously tested for activity against viruses, and extracts of *Cynometra cloiselii, Cynometra madagascariensis, Evonymopsis longipes, Ravensara retusa*, and *Terminalia seyrigii* were able to completely inactivate an HSV test inoculum at low concentrations [158]. During the COVID-19 pandemic, Madagascar has been developing COVID-Organics clinical trials with a herbal background [159].

#### 2.2.4. Antiviral Alkaloids

Another group of secondary metabolites with high antiviral potential against coronaviruses are the alkaloids. The bisbenzylisoquinoline alkaloids, tetrandrine, fangchinoline, and cepharanthine from roots of *Stephania tetrandra* inhibit expression of the human coronavirus, HCoV-OC43, spike and nucleocapsid proteins. These alkaloids were also able to provide beneficial immunomodulation [114,115] and were active against laboratory HIV-1 strains [160]. The isolated alkaloid lycorine from *Lycoris radiata* exhibited significantly greater inhibition of SARS-CoV (BJ-001) compared to the total alkaloid extract. However, lycorine extracted from plants showed lower inhibition of SARS-CoV (BJ-001) than synthetic lycorine [161]. Extracts of *Houttuynia cordata* L. contained the alkaloids arisolactam, piperolactam A, and caldensin, the terpene cycloart-25-ene-3b,24-diol, and several flavonoids [162]. A water extract of *Houttuynia cordata* significantly stimulated the proliferation of mouse splenic lymphocytes in a dose-dependent manner. *Houttuynia cordata* extracts have been shown to halt viral tRNA polymerase activity (RdRp) and increase secretion of the interleukins IL-2 and IL-10 [69]. These effects may be attributed to a synergistic interaction of specific compounds from the terpenoid, alkaloid, and flavonoid groups or the effect of their interaction in the plant water extract. The beneficial presence of natural compounds in the lungs after oral administration was further shown in the case of isoquinoline alkaloid emetin, which antagonizes viral replication, including that of coronaviruses. This secondary metabolite is found in the roots of the Brazilian shrub *Carapichea ipecacuanha* (*Rubiaceae*), and is processed into ipecac syrup [162,163]. As in the case of *Artemisia*, some antimalarial medicines have also been recommended for the treatment of COVID-19, namely chloroquine and hydroxychloroquine, which were synthesized as substitutes for quinoline alkaloid quinine that is extracted from the bark of the Peruvian tree, *Cinchona officinalis* [163,164]. The anecdotal and unproven use of *Artemisia* for COVID-19 following claims from politicians and others in low-income countries highlights the need for hard data to establish the active ingredients, especially to develop formulations and dosing, and to evaluate efficacy through controlled trials [164]. Another Brazilian herb, *Aspidosperma tomentosum*, inhibited replication of the avian metapneumovirus by 99% after the virus entered cells [42]. Plants belonging to the genus *Aspidosperma*, a member of the family *Apocynaceae,* are rich sources of β-carboline alkaloids, which make them potentially poisonous. Some of the identified alkaloids possess antitumor, antiplasmodial, antimicrobial, and antiviral activity [42,43,164]. The population of Valle of Juruena (Brazil) makes use of a wide array of medicinal plants, including *Aspidosperma tomentosum* extracts, in the treatments of respiratory ailments [165].

### 2.3. VOSviewer Visualization of Scientific Data Analysis of Antiviral Potential of Natural Compounds from Various Medicinal Herbs and Crops

With the rapid development and expansion of the field of plant-derived natural products and its global importance, it was worthwhile to conduct a timely assessment of the most influential articles (as measured by citations) and to identify seminal papers and scientific trends. To do this, we prepared a bibliometric network of publications dealing with antiviral compounds of plant origin using the software VOSviewer version 1.6.15 [166], a tool for the bibliometric analysis and visualization of scientific literature data [167]; the software is especially valuable for displaying large bibliometric maps in an easy-to-interpret way, and has been previously used to identify and analyze antimicrobial resistance [168], global research on leishmaniasis [169], and the 100 most-cited papers on the topic of nutraceuticals and functional foods [170], among others. The produced bibliometric network consists of nodes and edges that are weighted based on the frequency of the terms and the strength of the relationships between them; such analysis can provide critical insights into the mechanisms, hazards, and potential antiviral efficacy of plant extracts and isolated compounds as well as changes in research topics with respect to time. For our visualization, we used data from 173 papers published between 1967 and 2020, extracted from the Web of Science database. We selected all publications dealing with plant-based antiviral products present in any of the species presented in Table 1. For the complete list of papers, as well as the complete set of queries needed for data extraction, data used for all publications included the title, year of publication, authors’ names, nationalities, affiliations, name of publishing journal, keywords, abstract, times of citation, country, and H-index. In the following map, nodes correspond to paper keywords, and the edges indicate not only the relationship between two nodes, but also the strength of this relationship. Additionally, Figure 1 uses a color code for displaying the time of publication.

The visualization corresponds to a distance-based map, in which the space between items reflects the strength of the relationship between them. A smaller distance indicates a stronger relationship, making it easier to identify clusters of related elements. Likewise, the relative importance is indicated by the size of each circle. The output includes all the results obtained from all species mentioned in Table 1. In total, 11 clusters and 223 links were identified.

Our analysis shows that, as expected, the interest in the study of natural compounds for the treatment of diseases or developing plant-based drugs seems to increase when there are widespread health crises; in the graph, nodes for Chikungunya (the middle left) and H1N1 (bottom) are visible in connection with the main node of “antiviral activity”, and the graph also shows that the publications dealing with Chikungunya appear to be more common after 2014—similarly, most papers using the keyword H1N1 were published around 2010. The increase in scientific output in these topics appears to be related to both the Chikungunya outbreak from 2013 to 2017 and the H1N1 epidemic in 2009. According to the World Health Organization [171], the first documented outbreak of Chikungunya occurred in 2013, with autochthonous transmission in the Americas; in 2014, Europe faced its highest Chikungunya burden, and more than 1 million suspected cases were reported to the Pan American Health Organization (PAHO) regional office. In 2017, as in previous years, Asia and the Americas were the regions most affected by the Chikungunya virus, and the efforts to find a specific antiviral drug treatment redoubled. There is no specific antiviral drug treatment for Chikungunya, and the clinical management targets primarily the relieving of the symptoms; therefore, promising approaches such as the green synthesis or photosynthesis of AgNPs from medicinal plants [172] could allow a broader understanding of all the alternatives for treatment against viral diseases which have no specific antiviral or vaccines available. Additionally, the visual output shows a strong cooccurrence of the topics “antiviral” activity and “ethnopharmacobotany”, meaning that a good portion of the articles published on these topics are based on the traditional knowledge and the use of plants for medicine, namely, antivirals, antifungal or antibacterial, with most papers focusing on Chinese ethnopharmacobotany, as well as Australian aboriginal medicine and Indian and Korean traditional treatments, amongst others. This also reflects the importance of traditional knowledge in the pharmaceutical and medical research, as highlighted by refs [172,173]. Finally, the map also shows a wide variety of topics, with the most researched topics of the use of the selected medicinal plants and crops, including investigations on herpes simplex virus (type 1 and type 2), respiratory syncytial virus, flavonoids, influenza virus, and hepatitis virus (types a, b, and c). Curiously, the research interest regarding potential use of plants also includes the analysis of viruses exclusively infecting livestock, such as the porcine reproductive and respiratory syndrome virus (PRRS) or Aujeszky’s disease (pseudorabies) virus that also primarily infects pigs. Some of the investigations on animal infections could have applications for human health. This is the case for the Newcastle’s disease virus (NDV) (upper right in bright green), which causes a deadly infection in many kinds of birds and some mild flu-like symptoms in humans. It is important to mention that some of the compounds and plants identified by the queries also showed activities against nonviral diseases, such as the anticancer applications of plants of the *Plantago* genus (*Plantaginaceae*). The analysis showed that the dynamics of the research of the plant-based antiviral products are probably heavily influenced by current events of public health at the local and global scale, additionally suggesting a close relationship between ethnobotanical research and antiviral properties in plants; we suggest that these trends might be investigated further to deepen our understanding of the dynamics of available products.

## 3. Methodology

The available information on the medicinal herbs and crops which characterized antiviral potential was collected from scientific databases and covered from 1967 up to 2021. The following electronic databases were used: PubMed, Science Direct, Scopus, Web of Science, and Google Scholar. The search terms used for this review included coronaviruses group; SARS-CoV; biological active compounds; plant chemo diversity; antiviral potential; natural antiviral plant extracts; and plant-based antiviral drugs. No limitations were set for languages. Knowledge databases were combined with some of the primary traditional Azerbaijani manuscripts on herbal medicine. A total of 173 articles were included in the present review. A VOSviewer visualization of scientific data analysis of antiviral potential of natural compounds from various medicinal herbs and crops has been used. The tool for bibliometric analysis and visualization of scientific literature data was the software VOSviewer version 1.6.15.

## 4. Conclusions

The literature analysis of plants with significant antiviral potential was performed through a thorough literature search and VOSviewer visualization. A total of 66 medicinal herbs and crops with different origin native areas which characterized antiviral potentials were described. The most promising compounds to develop plant-based drugs can be reccomended the kaempferol glycosides, the scutellarein, baicalin and quercetin flavonoids; the saikosaponins triterpenoids; the lycorine, tetrandrine, fangchinoline, and cepharanthine alkaloids; the triterpene oleanane; and the terpene cycloart-25-ene-3b,24-diol. Even though preclinical studies suggest the potential effectiveness of described compounds to mitigate the current COVID-19 pandemic, clinical evidence is still missing. The dynamics of the worldwide research of the plant-based antiviral products are probably heavily influenced by current events of public health at the local and global scale, also suggesting a close relationship between ethnobotanical research and antiviral properties in plants which may be used for further studies to develop plant-based drugs against coronavirus, or to support backgrounds for healthy food recommendations during a pandemic.

## Figures and Tables

**Figure 1 molecules-26-00727-f001:**
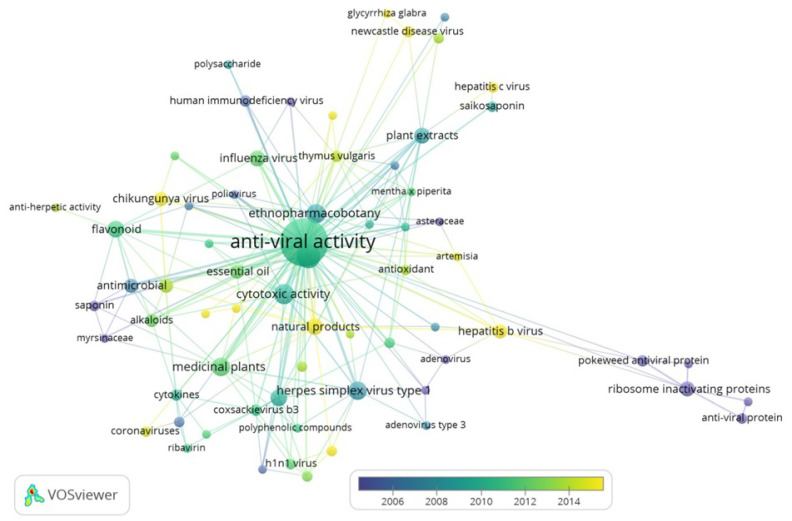
Schematic visualization of analyzed scientific data regarding the antiviral potential of natural plant resources.

**Table 1 molecules-26-00727-t001:** Description of medicinal plants and crops and their antiviral activity. The abbreviations of the authors of the scientific names have been omitted for the sake of clarity (see http://www.theplantlist.org).

Plant Species	Family	Plant Part	Origin Native Area	Mode of Action Pharmaceutical Activity	Reference
*Acacia nilotica*	Fabaceae	Whole plant	Africa and Middle East, Indian subcontinent	Inhibits human immunodeficiency virus (HIV) protease; antiviral and cytotoxic	[6,29]
*Alhagi maurorum*	Fabaceae	Gum tragacanth	South-east Europe, south-west Asia	Inhibits influenza and cold viruses; relieves cough, pectoral aches, fever, vomiting and thirst	[30]
*Allium sativum*	Alliaceae	Bulb	Central Asia, Iran	Inhibits avian coronavirus; antiviral, fungistatic	[31,32]
*Althaea officinalis*	Malvaceae	Whole plant	Western palearctic, boreal area, Europe, Asia and Africa	Anti-inflammatory in diseases of the upper respiratory tract; antitussive, chest emollient, immuno-modulator, antiviral	[33,34,35]
*Ampelozizyphus amazonicus*	Rhamnaceae	Whole plant	South America	Immunomodulation, anti-inflammatory	[36]
*Andrographis paniculata*	Acanthaceae	Leaves	India, Sri Lanka	Antiviral	[6,37]
*Anthemis hyalina*	Asteraceae	Whole plant	Mediterranean region, south-west Asia to Iran	Inhibits coronavirus replication and expression of transient receptor potential gene family	[38]
*Arrabidaea samydoides*	Bignoniaceae	Whole plant	South America	Antiviral activity against human herpes simplex virus-1 (HSV-1), vaccinia virus and murine encephalomyocarditis virus	[39]
*Artemisia sp. (Artemisia absinthium)*	Asteraceae	Whole plant	Eurasia, north Africa, North America	Reduces coronavirus replication; antibacterial, anti-inflammatory	[40,41]
*Aspidosperma sp.*	Apocynaceae	Whole plant	South America	Antiviral activity against avian metapneumovirus and other groups	[42,43]
*Bryophyllum pinnatum*	Crassulaceae	Whole plant	Madagascar	Anti-inflammatory immunomodulator; induces production of host antiviral agents; prescribed for polio and enteroviruses	[33,44]
*Camellia japonica*	Theaceae	Whole plant, Flowers	East Asia	High antiviral activity on porcine epidemic diarrhea virus (PEDV) of coronavirus family; inhibitory effects on key gene and protein syntheses during PEDV replication	[45,46]
*Cichorium intybus*	Asteraceae	Whole plant, Roots	Eurasia, Mediterranean region	Immunomodulation; antiviral activity against HSV-1 and adenovirus type 5	[37,47]
*Cinnamomum cassia*	Lauraceae	Bark	Vietnam and eastern Himalayas, China	Antiviral, anti-inflammatory; inhibits attachment of human respiratory syncytial virus	[48,49,50]
*Citrus trifoliata*	Rutaceae	Seeds	Northern China and Korea	Antiviral against oseltamivir-resistant influenza virus	[51]
*Clitoria ternatea*	Fabaceae	Whole plant	Indian sub-continent, Southeast Asia	Antiviral	[6]
*Cynara scolymus*	Asteraceae	Flower heads	Mediterranean region	ACE inhibitor, antiviral	[6,52]
*Desmodium canadense*	Fabaceae	Whole plant	North America	High antiviral activity towards coronaviruses	[53]
*Echinacea angustifolia*	Asteraceae	Flowers	North America	Antiviral activity against cold and flu viruses; inhibits viral growth and secretion of pro-inflammatory cytokines.	[54]
*Echinops sp.*	Asteraceae	Trehala manna	Iran	Antiviral, cough suppressant	[55]
*Echium amoenum*	Boraginaceae	Flowers	Iran, Caucasus, Russia	Antiviral	[56,57]
*Euphorbia sp.*	Euphorbiaceae	Roots	Southern Africa and Madagascar, North and South America, Mediterranean region	Antiviral activity against HIV-1, HIV-2, HSV-2 and SIVmac251	[35,58,59]
*Ferula assa-foetida*	Apiaceae	Oleo-Gum-resin	Iran, Afghanistan	Antiviral activity; great potency against H1N1; anti-inflammatory	[60,61]
*Firmiana simplex*	Malvaceae	Leaves	South Japan, China and Indonesia	Immunomodulation; general tonic and adaptogenic drug	[33]
*Glycyrrhiza glabra*	Fabaceae	Roots	Mediterranean area, Iran-Turan, Azerbaijan	Immunomodulation; antiviral activity against HSV-1, Epstein–Barr virus, human cytomegalo-virus, and RNA viruses such as influenza A, H5N1, and H1N1	[35,62,63,64]
*Gymnema sylvestre*	Apocynaceae	Leaves, Whole plant	Asia, Africa, Australia	Inhibition of viral DNA synthesis; immunomodulation	[6,65]
*Hippophae rhamnoides*	Elaeagnaceae	Fruits	Cold-temperate regions of Europe and Asia	Anti-influenza activities against influenza A/Victoria virus and B | Immunomodulation	[66,67,68]
*Houttuynia cordata*	Saururaceae	Whole plant	Southern Asia	Inhibits viral SARS-3CLpro and tRNA polymerase activity (RdRp); stimulates secretion of IL-2 and IL-10	[69]
*Humulus lupulus*	Cannabaceae	Inflorescences	Europe, western Asia, North America	Immunomodulation; antiviral activity against influenza and cold viruses, hepatitis C, and herpesvirus; inhibits viral replication	[70,71,72]
*Hyoscyamus niger*	Solanaceae	Whole plant	Continental Europe, Asia, Middle East	Viral inhibition; bronchodilator; antiviral effect against human influenza virus A/WSN/33	[6,73,74]
*Hypericum connatum*	Hypericaceae	Whole plant	North America, eastern Asia	High antiviral activity	[75]
*Inula helenium*	Asteraceae	Rhizomes, Roots	Eastern Europe, Caucasus, western Siberia, Far East and Central Asia	Anti-inflammatory	[71]
*Isatis tinctoria*	Brassicaceae	Roots extracts	Caucasus, Central Asia, eastern Siberia, western Asia	Inhibits cleavage activity of SARS-3CLpro enzyme; high antioxidant potential and anti-inflammatory effects	[76,77]
*Juniperus communis*	Cupressaceae	Fruits	North America, Europe, Asia	Inhibits replication, 3CLpro; anti-inflammatory, antiseptic	[78,79]
*Litchi chinensis*	Sapindaceae	Seeds	Southeastern China	Inhibit SARS-3CLpro; terpenoids inhibit HIV-1 protease	[17,53,80,81,82]
*Mentha piperita*	Lamiaceae	Whole plant	Europe, Middle East	High antiviral activity against coronavirus group	[17]
*Mosla sp.*	Lamiaceae	Whole plant	Eastern and south-eastern Asia, Himalayas	Anti-influenza activity	[83,84]
*Nigella sativa*	Ranunculaceae	Whole plant	Eastern Mediterranean, northern Africa, Indian Subcontinent, western Asia	Immunomodulator, anti-inflammatory agent, and broncho-dilator; antiviral activity against avian influenza virus (H9N2)	[85,86,87,88]
*Ocimum kilimandscharicum*	Lamiaceae	Whole plant	Central Africa, Southeast Asia	Antiviral activity against HIV-1, SARS-CoV-2	[6,21]
*Oplopanax elatus*	Araliaceae	Whole plant	North America, north-eastern Asia	Immunomodulation and anti-inflammatory activities	[33,89,90]
*Origanum vulgare*	Lamiaceae	Leaves, Stems	Western and south-western Eurasia, Mediterranean region	Respiratory and antiviral activity	[91,92,93]
*Pelargonium sidoides*	Geraniaceae	Leaves, Whole plant	South Africa	Decreases rhinovirus infection via modulation of viral binding proteins on human bronchial epithelial cells	[87,94]
*Plantago major*	Plantaginaceae	Leaves, Whole plant	Europe, Northern and central Asia	Anti-inflammatory; antiviral activity against herpesviruses and adenoviruses	[35,95,96]
*Punica granatum*	Lythraceae	Fruits, Peel, Seeds	Iran to northern India, Mediterranean region	Inhibits viral glycoproteins; antiviral activity against HSV-1 and influenza virus	[6,97]
*Rhaponticum carthamoides*	Asteraceae	Roots	Southern Siberia, Kazakhstan, Altay region	Immunomodulation	[33]
*Rosmarinus officinalis*	Lamiaceae	Whole plant	Mediterranean region	Antiviral activity against human respiratory syncytial virus; immunomodu-lator; anti-inflammatory	[33,98]
*Rubus sp.*	Rosaceae	Fruits, Flowers	Forest-steppe zones of Eurasia	Antiviral effect against human influenza virus	[33,71,99,100]
*Rhus coriaria*	Anacardiaceae	Fruit	Mild Mediterranean climates of southern Europe and western Asia	Antiviral potential	[101,102]
*Rosa sp.*	Rosaceae	Completely matured fruits	Europe, North America, Northwestern Africa	Immunomodulatory effects; antiviral activity against HIV and HSV	[103,104,105,106]
*Salvia officinalis*	Lamiaceae	Whole plant	Mediterranean basin	High binding to COVID-19 proteases; Inhibits SARS-CoV and HSV-1 replication	[21,107]
*Sambucus nigra*	Adoxaceae	Whole plant	Europe and North America	Antiviral activity against HIV, HSV, influenza, hepatitis, and coxsackievirus	[75,108]
*Saposhnikovia divaricata*	Apiaceae	Whole plant	China	High antiviral activity against PEDV corona-virus	[45,109]
*Scutellaria baicalensis*	Lamiaceae	Roots	China, Korea, Mongolia, Russian far east, Siberia	Inhibit nsP13 by affecting the ATPase activity	[110,111]
*Sphaeranthus indicus*	Asteraceae	Whole plant	Northern Australia, Indomalayan realm	Antiviral activity against mouse coronavirus; bronchodilation and anti-inflammatory activities	[6,112]
*Stachys schtschegleevii*	Lamiaceae	Leaves	Iran	Antiviral activity against SARS-CoV-2; anti-inflammatory potential	[113]
*Stephania tetrandra*	Menispermaceae	Roots	China, Taiwan	Inhibits expression of HCoV-OC43 spike and nucleocapsid proteins; immunomodulation and anticancer potential	[114,115]
*Strobilanthes cusia*	Acanthaceae	Leaves, Whole plant	Tropical Asia, Madagascar	Inhibits HCoV-NL63 via tryptanthrin; anti-influenza virus activity; anti-inflammatory potential	[6,116,117]
*Tabebuia sp.*	Bignoniaceae	Whole plant	South America	Antiviral potential	[118,119]
*Thymus vulgaris*	Lamiaceae	Whole plant	Southern Europe	High antiviral activity towards coronaviruses; antioxidant effects	[21,53]
*Thuja occidentalis*	Cupressaceae	Leaves Whole plant	Eastern Canada, north, central and upper north-eastern United States	Immunostimulation; antiviral activity against acute common cold	[120]
*Urtica dioica*	Urticaceae	Leaves	Europe, temperate Asia, and western North Africa	Inhibition of SARS coronavirus replication	[121]
*Viburnum opulus*	Adoxaceae	Fruits	Eastern Europe, Caucasus, western and eastern Siberia and Central Asia	Immunomodulation; anti-inflammatory effects	[33,122]
*Vitex trifolia*	Verbenaceae	Whole plant	Tropical East Africa, French Polynesia	Anti-inflammatory effects on lungs; immunomodulation; strong antiviral activity against HSV and mouse coronavirus (surrogate for human SARS virus)	[6,123]
*Zingiber officinale*	Zingiberaceae	Rhizome	Maritime Southeast Asia	Antiviral activity against human respiratory syncytial virus	[124]
*Ziziphus jujuba*	Rhamnaceae	Fruit	Southeastern Europe to China	Antiviral activity; potential therapeutic agent for treating influenza	[125]
*Zostera marina*	Zosteraceae	Whole plant	North America, Europe, Asia	Antiviral activity against influenza A virus	[126]

## Data Availability

The data presented in this study are available in this article.

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
