# Peer review of "COVID-19 Prophylaxis Efforts Based on Natural Antiviral Plant Extracts and Their Compounds"

_molecules, 2021, doi:10.3390/molecules26030727_

Round 1

Reviewer 1 Report

This paper presents the results of a review to identify the most popularly used Antiviral Plants during the COVID-19 pandemic. Even new information is identified, however, the findings is weakened by several issues including incomplete description of background, method and results, inconsistency of gap identified, results and discussion, and issues in the logical presentation of the study aims.  Details relating to these and other issues are presented below.

Title and abstract:

  1. Suggest: revise the title and abstract part: Both does not match research purpose and content.

Introduction

  1. In general, Suggest to:

Make it brief.

Provide the necessary background information only.

Provide the gap of this study

  1. Authors should revise introduction section: appropriate paragraphs using topic sentences.

Suggest: revise the aim of this study AND describe the background related to the topic or change the topic.

  1. Authors should provide the PICO on medicinal herbs in the context of global health system / health market/ academic research / methodological issues.
  2. Authors should state the reasons for their study in relation to comparing the research methods and antiviral plants including and reviewing previous literature or observation. Additionally, authors should revise the last sentence…aims of this study...
  3. suggest to provide the gap of this study from previous researches

Methods

Suggest to revise the methodology 

- Authors should include more articles related to medicinal plants use.

- provide search strategies for this DRAFT.

- provide how to conduct “the comprehensive bibliometric analysis” (how to analyse the document using any methods.)

Results and Discussion

- the findings more logically and critically and so what? 

- Suggest to provide discuss section

Conclusion

Suggest to revise the conclusion

-  Clearly state the answer to the main research question.

- Summarize and reflect on the research.

- suggest recommendations for future work on the topic.

- Indication what new knowledge this study has contributed.

Author Response

Dear Reviewer, first of all thank you for the very important comments. Please, see answers on the your comments bellow.

Reviewer 1

This paper presents the results of a review to identify the most popularly used Antiviral Plants during the COVID-19 pandemic. Even new information is identified; however, the findings is weakened by several issues including incomplete description of background, method and results, inconsistency of gap identified, results and discussion, and issues in the logical presentation of the study aims.  Details relating to these and other issues are presented below.

Title and abstract:

  1. Suggest: revise the title and abstract part: Both does not match research purpose and content.

Thank you for the comment. We fully agree with you. The title and abstract were revised regarding information presented in the article. Please, see changes in a red color.

Introduction

  1. In general, Suggest to:

Make it brief.

Provide the necessary background information only.

Provide the gap of this study

Thank you, the all your suggestions. The Introduction part was updated. It is briefer with just necessary background information and missed gaps. Please, see changes in a red color. The information about vaccine development was added. The possible role of plant-based products in it has been described with original references.

  1. Authors should revise introduction section: appropriate paragraphs using topic sentences.

Suggest: revise the aim of this study AND describe the background related to the topic or change the topic.

  1. Authors should provide the PICO on medicinal herbs in the context of global health system / health market/ academic research / methodological issues.
  2. Authors should state the reasons for their study in relation to comparing the research methods and antiviral plants including and reviewing previous literature or observation. Additionally, authors should revise the last sentence…aims of this study...
  3. suggest to provide the gap of this study from previous researches

Thank you for the comment. The Introduction was revised. It was stated correct aim regarding presented study. It was provided information about medicinal herbs in the context of global health system / health market/ academic research / methodological issues.

Methods

Suggest to revise the methodology 

- Authors should include more articles related to medicinal plants use.

Thank you. Its done. The more article regarding medicinal plants use included.

- provide search strategies for this DRAFT.

Thank you for the comment. The literature analysis which was done to create a scientific database of plants with antiviral potential (Table 1), obtained results from a search of PubMed, Scopus and Web of knowledges databases combined with some of the primary traditional Azerbaijan manuscripts on herbal medicine.

- provide how to conduct “the comprehensive bibliometric analysis” (how to analyse the document using any methods.)

Thank you for the comment. The “bibliometric analysis was removed from conclusion part. The description of methodology “The literature analysis which was done to create a scientific database of plants with antiviral potential (Table 1), obtained results from a search of PubMed, Scopus and Web of knowledges databases combined with some of the primary traditional Azerbaijan manuscripts on herbal medicine.” Was added and significantly improve text of paper.

Results and Discussion

- the findings more logically and critically and so what? 

- Suggest to provide discuss section

 Thank you for the suggestion. In the text connected with plant natural compounds detailed analysis is also present discussion part. The all updates in the text in a red color.

Conclusion

Suggest to revise the conclusion

-  Clearly state the answer to the main research question.

- Summarize and reflect on the research.

- suggest recommendations for future work on the topic.

- Indication what new knowledge this study has contributed.

Thank you for the suggestions and comments. The Conclusion part was rewritten with clear the answer to the main research question. The results summarization was rewritten. The changes in a red color.

Reviewer 2 Report

I found this an interesting and useful read. In the present circumstances of course almost anything of relevance to covid is likely to be of interest! The review is strong on generalities and less so on specifics. Nonetheless I think that it is a useful compilation of background and references for anyone getting interested or involved in the field and worth publishing. A few comments to consider areas follows.

  1. Obviously since the article was written, vaccines have come on stream and vaccination programmes are now being enacted. It would be worth mentioning this, with the caveat of course that only time will tell how effective and robust these programmes are likely to be, thus necessitating  alternative strategies to continue to be pursued.
  2. the section lines 153-174 is rather poorly written and needs revising to bring it into line with the rest of the article.
  3. the section headed "antiviral  terpenoids" lines 230 onwards makes extensive references to non-terpenoid plant products and repeats other sections. This needs to be rationalised and rewritten.

Author Response

Dear Reviewer, thank you for your opinion and very useful comments and suggestions. Please, see responses bellow.

I found this an interesting and useful read. In the present circumstances of course almost anything of relevance to covid is likely to be of interest! The review is strong on generalities and less so on specifics. Nonetheless I think that it is a useful compilation of background and references for anyone getting interested or involved in the field and worth publishing. A few comments to consider areas follows.

  1. Obviously since the article was written, vaccines have come on stream and vaccination programmes are now being enacted. It would be worth mentioning this, with the caveat of course that only time will tell how effective and robust these programmes are likely to be, thus necessitating  alternative strategies to continue to be pursued.

Thank you for the comment and suggestion. Novel information about vaccines development and possible role of plant-based products in it has been added in Introduction part. Please, see changes in a red color.

  1. the section lines 153-174 is rather poorly written and needs revising to bring it into line with the rest of the article.

Thank you for the comment. the section lines 153-174 was rewritten and updated with current information. Please, see changes in a red color.

  1. the section headed "antiviral  terpenoids" lines 230 onwards makes extensive references to non-terpenoid plant products and repeats other sections. This needs to be rationalised and rewritten.

Dear Reviewer, thank you for the comments. The text was rewritten but salikosaponins is triterpenoid glycoside and present a class of triterpene saponins. Please, see reference below

 Saikosaponins are a class of triterpene saponins (Yu YH, Xie W, Bao Y, Li HM, Hu SJ, Xing JL. Saikosaponin a mediates the anticonvulsant properties in the HNC models of AE and SE by inhibiting NMDA receptor current and persistent sodium current. PLoS One. 2012;7(11):e50694. doi: 10.1371/journal.pone.0050694. Epub 2012 Nov 29. PMID: 23209812; PMCID: PMC3510157.)

Saikosaponin-A, a triterpenoid glycoside (Park KH, Park J, Koh D, Lim Y. Effect of saikosaponin-A, a triterpenoid glycoside, isolated from Bupleurum falcatum on experimental allergic asthma. Phytother Res. 2002 Jun;16(4):359-63. doi: 10.1002/ptr.903. PMID: 12112293.)

Round 2

Reviewer 1 Report

Thank you for addressing the comments. The draft is improved but still there are several points that needs to be addressed as stated below.

Title

  1. Suggest: revise the title: When considering the abstract, introduction, and conclusion, the tile is required to be changed.

- suggest to narrow down the focus of this study.

Introduction

  1. Authors should revise introduction section:

- suggest to revise or remove first and second paragraph which is not relevant contents for this study.

- suggest to stick to “anti-viral potential of plant extracts” or “anti-viral potential of natural product”

  1. Result and discussion

- Authors should revise the result and discussion section: considering appropriate existing and updated articles related to “anti-viral potential of plant extracts” or “anti-viral potential of natural product” use among general population or whom.

  1. Conclusion

Suggest to revise the conclusion

-  Clearly state the answer to the main research question.

- Summarize and reflect on the research.

- Indication what new knowledge this study has contributed.

(end)

Author Response

Dear Reviewer,

 Thank you very much for your comments and suggestions. Please, see answers for the comments bellow. The text of MS with updated text and added literature references in the attachment.

Title

  1. Suggest: revise the title: When considering the abstract, introduction, and conclusion, the tile is required to be changed.

- suggest to narrow down the focus of this study.

Thank you for the comment. The title was updated with more focus on presented study. Please, see changes in a red color.

Introduction

  1. Authors should revise introduction section:

- suggest to revise or remove first and second paragraph which is not relevant contents for this study.

- suggest to stick to “anti-viral potential of plant extracts” or “anti-viral potential of natural product”

Thank you for the suggestions. The first and second paragraph were combained together. Some text was removed. As anti-viral plant extracts described were connected with COVID-19 pandemy nowadays so we would like to keep some current information of it in the Introduction. It is fully in line with title, abstract, introduction, conclusion.

  1. Result and discussion

- Authors should revise the result and discussion section: considering appropriate existing and updated articles related to “anti-viral potential of plant extracts” or “anti-viral potential of natural product” use among general population or whom.

Thank you for the comment. The updated articles regarding “anti-viral potential of natural product” use among general population were added and text was updated. Please, see changes in a red color.

  1. Conclusion Suggest to revise the conclusion

-  Clearly state the answer to the main research question.

- Summarize and reflect on the research.

- Indication what new knowledge this study has contributed.

(end)

Thank you for the suggestion. The Conclusion part was updated with clearly state the answer to the main research question. Please, see updated version in a red color.